# Preharvest Reduction in Nutrient Solution Supply of Pepper (*Capsicum annuum* L.) Contributes to Improve Fruit Quality and Fertilizer Efficiency While Stabilising Yields

**Junzheng Wang** [1,2,3]**, Zixing Gao** [1,2,3]**, Tao Sun** [1,2]**, Wenxian Huang** [1,2]**, Yuanjie Jia** [1,2,3]**, Xiaojing Li** [1,2,3]**, Zhi Zhang** [1,2,3,*] **and Xiaohui Hu** [1,2,3,*]

1   College of Horticulture, Northwest A&F University, Xianyang 712100, China
2   Key Laboratory of Protected Horticultural Engineering in Northwestern China, Ministry of Agriculture and Rural Affairs, Xianyang 712100, China
3   Shaanxi Protected Agriculture Research Centre, Xianyang 712100, China
*   Correspondence: zhangzhione@126.com (Z.Z.); hxh1977@nwafu.edu.cn (X.H.)

**Abstract:** Optimising fertilisation is an important part of maximising vegetable yield and quality whilst minimising environmental hazards. An accurate and efficient scheme of irrigation and fertiliser based on plants' nutrient requirements at different growth stages is essential for the effective intensive production of greenhouse pepper (*Capsicum annuum* L.). In this study, the effects of reducing fertilisation rate by 20%, 40%, 60% and 80% from the day 6 to day 0 before harvest for each layer of peppers on growth, yield, quality and nutrient utilisation were evaluated. The results showed that the morphological indicators (plant height and stem diameter) and biomass of plants decreased gradually with the increase in fertiliser reduction rate. Compared with control (CK) plants, the 20–40% reduction in fertiliser application rate did not cause a significant decrease in biomass and stem diameter but significantly increased the accumulation of N (13.52–15.73%), P (23.09% in 20% reducted-treatment) and K (13.22–14.21%) elements in plants. The 20–80% reduction in fertiliser application before harvest had no significant effects on the nutrient agronomic efficiency of N, P and K elements. However, it decreased the physiological nutrient efficiency and significantly improved the nutrient harvest index of N, P and K. Appropriate reduction in fertiliser application significantly increased the nutrient recovery efficiency (20–40% reduction) and nutrient partial-factor productivity (40% reduction) of N (3.35–6.00% and 12.87%), P (2.47–2.92% and 14.01%) and K (7.49–15.68% and 14.01%), respectively. Furthermore, reducing the fertilisation rate by 20–40% before each harvest had a certain positive effect on the C and N metabolism of pepper leaves and fruits. In particular, the activities of N metabolism-related enzymes (nitrate reductase, nitrite reductase, glutamine synthase, glutamate synthase and glutamate dehydrogenase) and C metabolism-related enzymes (sucrose phosphate synthase, sucrose synthetase, acid invertase and neutral invertase) in leaves and fruits did not significantly different or significantly increased compared with those in CK plants. The results of the representative aromatic substance contents in the fruit screened by the random forest model showed that compared with the CK plants, reducing the fertiliser application by 20–40% before harvest significantly increased the content of capsaicin and main flavour substances in the fruit on the basis of stable yield. In summary, in the process of pepper substrate cultivation, reducing the application of nutrients by 40% from the day 6 to day 0 before each harvest could result in stable yield and quality improvement of the pepper. These results have important implications for institutional precision fertilisation programs and the improvement of the agroecological environment.

**Keywords:** pepper; mineral deficiency stress; yield; fruit quality; nutrient use efficiency; nitrogen metabolism

## 1. Introduction

With the continuous increase in population and the diminution of usable space for agricultural production [1], the synergistic improvement of yield and quality of vegetables has become an inevitable requirement for agricultural producers, and it drives people to apply different agronomic measures to achieve this goal [2,3]. Therefore, various management practices, such as environmental control [4], pesticide application and irrigation and fertilisation management [5,6], are adopted by farmers. At present, irrigation and fertilisation is the primary input for improving the yield and quality of vegetable in agricultural production activities in China [7]. However, affected by the characteristics of intensive production, vegetable production systems are generally characterised by large applications of mineral fertiliser [8]. Commonly, the chemical fertiliser supply considerably exceeds crop element requirements, eventually leading to a series of problems, such as deterioration of soil quality, increased wastewater discharge and greenhouse gas emissions [9,10]. Healthy growth and development of vegetables need a balanced supply of water and fertiliser, and the ratio and amount of nutrient elements could affect vegetable yield and quality [11,12]. Thus, over the past decade, attention has been focused on the rapid development of a method for precise fertiliser applications in greenhouse crop management [13,14]. However, the dynamic vegetable growth process makes plants have different nutrient requirements at different growth stages [15]. Therefore, a precise application scheme of nutrients based on different growth stages of vegetables has become the key to achieving precise fertilisation at present.

Pepper (*Capsicum annuum* L.) is an important vegetable cultivated worldwide, being rich in piquancy, nutritional and pigment contents of fruits. In 2020, the global pepper planting area was about 2.07 million hectares, the global pepper production was about 36.14 million tons, and annual pepper production in China was 16.68 million tons (https://www.fao.org/faostat/en/#compare (accessed on 20 August 2022). Nowadays, the yield and quality of pepper fruit are getting attention from growers and consumers. Similar to tomatoes, pepper fruits contain many aspects, including appearance, taste, nutrients and characteristic aroma compounds [16]. As the main indicator of nutritional quality, the concentration of sugars, acids, phenols and minerals in pepper fruit determines its flavour and nutrition, of which capsaicin and aromatic substances have been regarded as important indicators for fruit flavour quality [17,18]. The yield and quality of pepper are determined by many factors. In a specific geographical environment, water and fertiliser management is undoubtedly the most agronomic measure to achieve different pepper varieties with stable yield and quality [19]. Some scholars have conducted several explorations on the precise management strategies of water and fertiliser for pepper [20–22]. However, a gap in the agricultural management strategy for implementing precise fertilisation by considering specific growth stages remains due to the different requirements of water and fertiliser in different growth stages of peppers and the diversity of species.

Organic substrate cultivation is an ecological and efficient cultivation model that reuses agricultural wastes and effectively solves the problem of soil environmental degradation, the application area of such cultivation method is increasing year by year [23]. Compared with soil cultivation, smaller rhizosphere space indicates that plants require a more precise supply of fertilisers for optimal cultivation and economic benefits. In addition, during the whole growth phase of the crop, the fertiliser requirement steadily increased with the increase in yield [24]. However, before each layer of fruit was harvested, the fertiliser required by the crop tended to briefly decrease [25]. Previous methods have mostly explored precise fertilisation strategies based on the overall amount of fertiliser during the whole growth period [26,27]. The process often ignored the changes in the physiological requirements of crops before ripening. Therefore, exploring precise fertilisation methods under substrate culture based on specific growth stages and coordinating the responses of different pepper indicators are crucial for guiding the production.

The author's previous study found that reducing nutrient solution supply 6 days before each harvest could maintain pepper yield and improve fruit quality [28]. However,

quantitative indicators of accurate reduction in nutrient solution amount before harvest, which is more suitable for pepper production, are still lacking. Based on the limitations in the current literature, the current study has the following aims: (1) to study the effect of reducing the application of different doses of fertiliser before harvest on pepper fruit quality, yield, nutrient and water use efficiency (WUE); (2) to explore the response of plant N and C metabolism to different nutrient solution reduction treatments; and (3) to determine the optimal nutrient solution reduction dose that could stabilise yield and improve quality through random forest machine learning model method.

## 2. Materials and Methods

### 2.1. Experimental Site and Materials

Experiments were carried out in the greenhouse base of Yangling (Shaanxi, 34°15′56′′ N, 108°03′40′′ E; 521 m above mean sea level) from 31 March to 6 August 2020. The daily average temperature and relative air humidity were 15.3 °C–32.0 °C and 29.9–95.0% during the cultivation, respectively (HOBO Micro Station and sensors, Onset Computer Company, Massachusetts, MA, USA). Before the start of the field experiments, pepper (*C. annuum* L. cv. Bolon RZ F1) seeds were grown to the five-leaf and one-centre stage in a greenhouse with a controlled environment. Afterwards, the seedlings were transplanted into substrate bags (length × width × height = 0.90 × 0.20 × 0.16 m). Each substrate bag contained two seedlings, and the 18 L substrate was composed of decomposed rotten cow dung, peat, mushroom residue waste and vermiculite (the volume ratio was 2:7:4:1, respectively). The substrate contained 334.21 mg·kg$^{-1}$ of nitrate nitrogen, 75.53 mg·kg$^{-1}$ of ammonium nitrogen, 629.54 mg·kg$^{-1}$ of available P, 940.31 mg·kg$^{-1}$ of available K, 42.92% of organic matter and 0.36 g·cm$^{-3}$ of bulk density at pH 6.65, with electrical conductivity (EC) of 1537 uS·cm$^{-1}$. The line spacing of the substrate bags was 0.80 m, the small line spacing was 0.40 m, and the plant spacing was 0.35 m.

### 2.2. Experimental Design and Field Management

With a one-strength Yamazaki nutrient solution as the base nutrient solution formula [29], the experiment was designed to reduce the rate of nutrient solution to different degrees before harvesting each layer of fruit. From 15 days after planting (15 April), the same level of the nutrient solution was applied in each treatment every 3 days and 300 mL·plant$^{-1}$ was irrigated each time. From day 33 after planting (3 May), it began to enter the experimental treatment stage. Each treatment was designed to apply the same dose of nutrient solution from day 15 to day 7 before harvesting each layer of fruit. Due to the differences in the nutrient requirements of plants at different harvesting periods, the application rate of nutrient solution before the first- to third-layer pepper harvesting was 500 mL·plant$^{-1}$ from the 15–7th day before each harvesting, whilst that for the fourth- to sixth-layer pepper harvesting was 1000 mL·plant$^{-1}$ [15,25]. However, from day 6 to day 0 day before each layer of pepper was harvested, different nutrient solution reduction rates were set for each treatment (Table 1). Within the experimental time range (from the 15th day before the first pepper harvest to the end of the experiment), the nutrient solution was applied every 2 days, eight times before each layer of fruit was harvested. The relative water content (RWC) of the substrate was controlled at 60% by irrigation during the whole growth period. The difference between the reduced nutrient solution supply and the normal water supply was supplemented after fertilisation to maintain the RWC of the substrate within the set range. A handheld matrix moisture meter (HH150, Delta-T Devices Ltd., UK) was used to determine the relative moisture content of the matrix. The experimental plots measured 6.5 m × 1.2 m. Each treatment was replicated three times, with a total of 15 plots. Each plot was planted with 38 peppers. After the experiment started, mature peppers were harvested every 15 days for a total of six times.

**Table 1.** Experimental factors and corresponding levels.

| Treatment | First Three Harvesting Periods (Each Plant Every Time) | | Last Three Harvesting Periods (Each Plant Every Time) | |
| --- | --- | --- | --- | --- |
| | 15–7 Days before Each Harvest | 6–0 Days before Each Harvest | 15–7 Days before Each Harvest | 6–0 Days before Each Harvest |
| CK | | 500 mL | | 1000 mL |
| T$_1$ | | 400 mL | | 800 mL |
| T$_2$ | 500 mL | 300 mL | 1000 mL | 600 mL |
| T$_3$ | | 200 mL | | 400 mL |
| T$_4$ | | 100 mL | | 200 mL |

*2.3. Measurements and Methods*

2.3.1. Growth and Yield

At the beginning of the experiment, eight plants were randomly selected from each treatment and marked for the determination of morphological indicators and the recording of fruit yield throughout the experiment period. Plant height and stem diameter were measured every 10 days for a total of five times. Plant height indicates the height from the base of the stem to the central growing point, and stem diameter indicates the diameter of the stem at the base. Each time the peppers were harvested, the weight of the fruit per plant in each treatment was weighed with an electronic balance, and the total yield of the plant was the total weight of the fruit harvested six times. After each harvest, the individual fruits were placed in an oven for drying to obtain the dry-matter accumulation of each plant (after drying was conducted at 105 °C for 30 min, it was conducted again at 70 °C to constant weight). Assays were carried out in three biological replicates.

2.3.2. Plant Biomass and Element Accumulation

The total dry biomass of a plant is the sum of the biomass of its roots, stems, leaves and fruits [30]. Amongst them, the total dry-matter accumulation of the fruits was the sum of the total dry-matter mass of each harvest. The naturally shed stems and leaves of the marked plants were collected to measure the corresponding dry biomass, which was then added to calculate the total dry biomass of the plants.

For mineral analysis, dried plant tissues (leaf, stem, fruit, and root) were ground separately in a Wiley mill to pass through a 50-mesh screen and then 0.5 g of the dried plant tissues were used to analyse the following macronutrients: N, P and K. After the plant tissues were mineralised with sulfuric acid, the concentration of N and K was measured with a fully automatic continuous flow analyser (AA3, SEAL Analytical, Hamburg, Germany). The K concentration was measured with a flame photometer (M410, Sherwood Company, Cambridge, England). The total N, K and P in the leaf, shoot, root and fruit were determined by following the method of Qu et al. [27]. Assays were carried out in six biological replicates.

2.3.3. Nutrition Quality of Fruit

During the third and fourth harvesting, 10 mature peppers with the same growth potential were selected from the same leaf position for each treatment, refrigerated in liquid nitrogen and then stored at −80 °C. The fruit samples collected twice were ground and mixed with liquid nitrogen for the determination of quality indicators. Soluble reducing sugar content was estimated in accordance with the method of Miller [31]. Soluble protein content was measured by Coomassie brilliant blue G-250 staining [32]. Soluble protein was determined in accordance with the accumulation of protein-pigment combinations at 595 nm. Vitamin C and organic acid were determined by following the method of Bona et al. [33]. Nitrate content was determined using the method of Monforte-Gonzalez et al. [34]. The soluble sugar content of pepper was measured in accordance with the anthranone-sulfuric acid assay described by Liu et al. [35]. The content of capsaicin was

determined using the method of Zhang et al. [36]. Assays were carried out in three biological replicates, with each replicate containing 10 fruits.

### 2.3.4. Flavour Quality of Fruit

The flavour qualities were analysed via gas chromatography–mass spectrometry (GC–MS), which was performed using an ISQ GC–MS combined instrument (Thermo Fisher Scientific, Waltham, MA, USA) equipped with HP-INNOWAX-fused silica capillary column (30 m × 0.25 mm, 0.25 um), and stirred samples using a constant-temperature magnetic stirrer (Troemner, Philadelphia, PA, USA). An SPME manual injection handle, solid-phase microextraction (Supelco, Bellefonte, PA, USA) and a homogeniser (Philips, Amsterdam, Holland) were also utilised. GC was performed as follows: firstly, 5 g of chili fruit homogenate was accurately and quickly weighed and placed into a 40 mL headspace vial containing 1 g of sodium chloride and a rotor. Meanwhile, a standard sample of 10 μL 2-octanone (0.1 μg·mL$^{-1}$) was added to each headspace bottle as an internal standard. The bottle was immediately covered by sealing it with tin foil and then placed in a constant-temperature magnetic stirrer (50 °C, 500 rpm) for 10 min. Headspace solid-phase microextraction was then used for adsorption for 40 min, and the GC gasification chamber was immediately inserted. The desorption time for GC–MS was 2.5 min. After the sample detection was completed, the samples were separated to form different peaks. The results of the identification were reported only when the matching degree and purity were 800 via computer retrieval and by comparing them with the standard mass spectrum of NIST 2011 in reference to the positive and negative matching degrees and those reported in the literature. The components were searched by NIST/Wiley. The volatile substances in the pepper fruits were quantitatively analysed by a standard internal method, the formula of which is as follows:

$$Z = \frac{\frac{S1}{S2} \times m \times 1000}{M}$$

where $Z$ represents the volatile matter content (μg·kg$^{-1}$); $S1$ and $S2$ denote the peak area of the sample and the peak area of the internal standard, respectively; and $m$ and $M$ indicate the mass of the internal standard (μg) and the mass of the sample (g), respectively. Assays were carried out in three biological replicates.

### 2.3.5. Fertiliser Use Efficiency (FUE) and WUE

The amount of irrigation and fertilisation throughout the experimental period was recorded for accurate calculation of *WUE* and nutrient use efficiency (*NUE*) as follows [37]:

$$NAE = \frac{TY_N - TY_0}{N_A}$$

$$NRE = \frac{TU_N - TU_0}{N_A}$$

$$NPFP = \frac{TY_N}{N_A}$$

$$NPE = \frac{TY_N - TY_0}{TU_N - TU_0}$$

$$NHI = \frac{G_N}{TU_N}$$

$$WUE = \frac{TY_N}{W_A}$$

where *NAE*, *NRE*, *NPFP* and *NPE* represent the agronomic efficiency, recovery efficiency, partial-factor productivity and physiological efficiency for the applied nutrient (N, P and K), respectively; NHI represents the nutrient harvest index; $TY_N$ and $TY_0$ are the pepper yields (kg·ha$^{-1}$) with and without nutrient application, respectively; $TU_N$ and $TU_0$ (kg·ha$^{-1}$) are

the amounts of plant N, P and K uptake with and without fertilisation, respectively; $N_A$ is the N, P and K application amount (kg ha$^{-1}$); $G_N$ is the fruit nutrient (N, P and K) uptake (kg·ha$^{-1}$) and $W_A$ represents the amount of irrigation during the experiment (m$^3$·hm$^{-2}$).

### 2.3.6. C and N Metabolism

At the peak fruiting period (the day of the fourth fruit harvest), ten leaves and eight fruits were collected from the same nodes in each treatment to determine the physiological indicators related to N and C metabolism. Leaf or fruit samples were ground and pulverised in liquid N by mortar and pestle. The activities of all N and C metabolism enzymes were determined by enzyme-linked immunosorbent assay (Fankew, Shanghai FANKEL Industrial Co., Ltd., Shanghai, China) in accordance with the manufacturer's instructions. The enzymes included nitrate reductase (NR, EC 1.6.6.1), nitrite reductase (NiR, EC1.7.2.1), glutamine synthetase (GS, EC 6.3.1.2), glutamate synthase (GOGAT, EC 1.4.1.14), glutamate dehydrogenase (GDH, EC 1.4.1.2), sucrose phosphate synthase (SPS, EC 2.4.1.14), sucrose synthetase (SS, EC 2.4.1.13), acid invertase (AI, EC 3.2.1.26) and neutral invertase (NI, EC 3.2.1.26). The assays were carried out in three biological replicates.

### 2.4. Data Analysis

All data were analysed with SPSS 23.0 statistical software package (IBM Corporation, Armonk, NY, USA) by using least significant difference (LSD) at a significance level of $p < 0.05$. Figures were drawn using GraphPad Prism 5 (V 5.0) (https://www.graphpad.com (accessed on 10 May 2019) and Microsoft Excel 2016. *R* language was employed to fit the random forest model. A correlation heatmap of the aromatic substances was also constructed.

## 3. Results

### 3.1. Morphological Characteristics of Pepper

Under the premise of consistent environmental conditions, changes in plant morphological development could intuitively reflect plant nutrient levels. On the 0th and 10th days after the start of the experiment, the plant height and stem diameter of each fertiliser reduction treatment did not significantly differ (Figure 1). With the increase in days, the growth rate of plants treated with a 40–80% reduction in chemical fertiliser before harvest (T2, T3 and T4 treatments) showed a decreasing trend, whilst the change ratio of plants treated with a 20% reduction (T1 treatment) was not significantly different from that of the control (CK) plants. On the 40th day after treatment, the plant heights of T2, T3 and T4 treatment plants were significantly reduced by 3.29–4.92% compared with that of the CK plants, and the stem diameter of the T4 treatment plants was significantly reduced by 8.07% compared with that of the CK plants (Figure 1a).

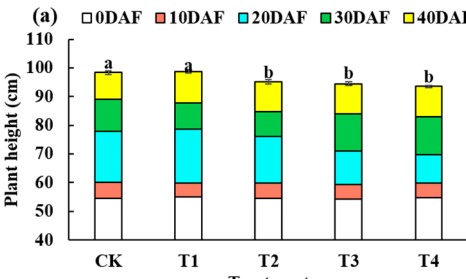
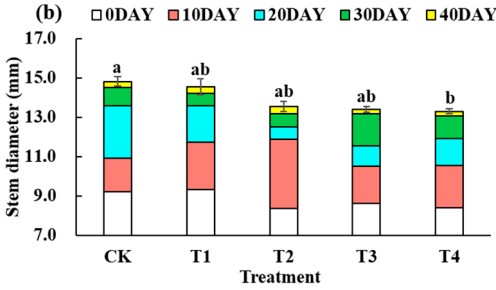

**Figure 1.** Effects of reducing fertiliser application before harvesting on plant height (**a**) and stem diameter (**b**). Data were the means of three replicates with standard error (±SE). Means followed by the same lowercase letter in the same column were not significantly different according to Tukey's multiple range test results at *p* of 0.05 (n = 3). With no fertiliser reduction as the control (CK), T1, T2, T3, and T4 treatments represented 20%, 40%, 60% and 80% reduction in fertiliser application from 0 to 6 days before each harvest, respectively. The 'DAF' indicates days after entering the fertilisation treatment.

### 3.2. Plant Biomass and Element Accumulation

The accumulations of plant biomass and element throughout the experimental period were measured, and the results showed that the reduction in fertiliser application to different degrees before harvesting each layer of pepper had no significant effect on the dry-matter accumulation of roots, stems and leaves (Figure 2). However, compared with that in CK, the fruit biomass in T3 and T4 treatments significantly decreased by 11.21–14.46%, whereas the fruit biomass of peppers treated with T1 and T2 was not significantly affected (Figure 2a). The biomass of the whole plant with different treatments showed the same trend as that of the fruit. The results of plant element accumulation during the entire experimental period showed that the accumulation of N, P and K elements (13.52%, 23.09% and 14.21%, respectively) in the whole plant treated with T1 significantly increased compared with that in CK (Figure 2b). The accumulation of N and K elements (15.73% and 13.22%) in plants treated with T2 also significantly increased compared with that in CK plants, whereas the element accumulation under T3 and T4 treatments were not significantly affected.

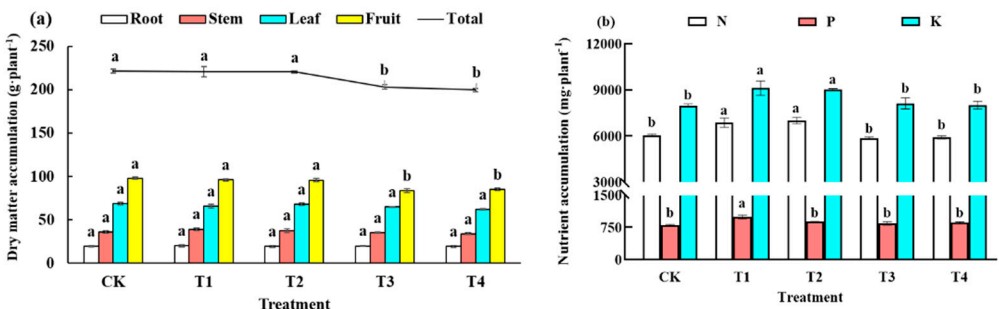

**Figure 2.** Effects of reduced fertiliser application before harvesting on plant biomass and element accumulation. (**a**) represents the dry-matter accumulation in the roots, stems, leaves and fruits of plants throughout the experimental period. (**b**) represents the total accumulation of N, P and K elements in plants during the experiment. Data were tested at the end of the experiment, and they were the means of six replicates with standard error (±SE). Means followed by the same lowercase letter in the same column were not significantly different according to Tukey's multiple range test results at *p* of 0.05 (n = 6). With no fertiliser reduction as the control (CK), the T1, T2, T3 and T4 treatments represented 20%, 40%, 60% and 80% reduction in fertiliser application from 0 to 6 days before each harvest, respectively.

### 3.3. Changes in Physiological Indices Caused by Reduced Fertiliser Application before Harvesting

3.3.1. Photosynthetic C Metabolism-Related Enzymatic Activities

Photosynthetic C metabolism is an important indicator for evaluating the photosynthetic and production capacities of plants, and it is closely related to the C metabolism process. Compared with the CK plants, reducing fertiliser application by 20–40% from 1 day to 6 days before each harvest significantly increased the NI activity in leaves by 91.69–105.19% (Figure 3). Reducing the application of fertiliser treatments also significantly increased the SPS activity in leaves by 20.79–24.22%, whereas it did not show a significant effect on the AI activity. Meanwhile, only the reduction in application rate by more than 40% significantly reduced the SS activity in leaves. In fruits, the NI and AI activities significantly increased by 17.77–21.68% and 22.84–34.72%, respectively, with a 20–40% reduction in fertiliser application before harvest. Unlike in leaves, the reduction in fertiliser application before harvest did not affect the SPS activity in fruit, but all four treatments significantly increased the SS activity (27.97–37.10%).

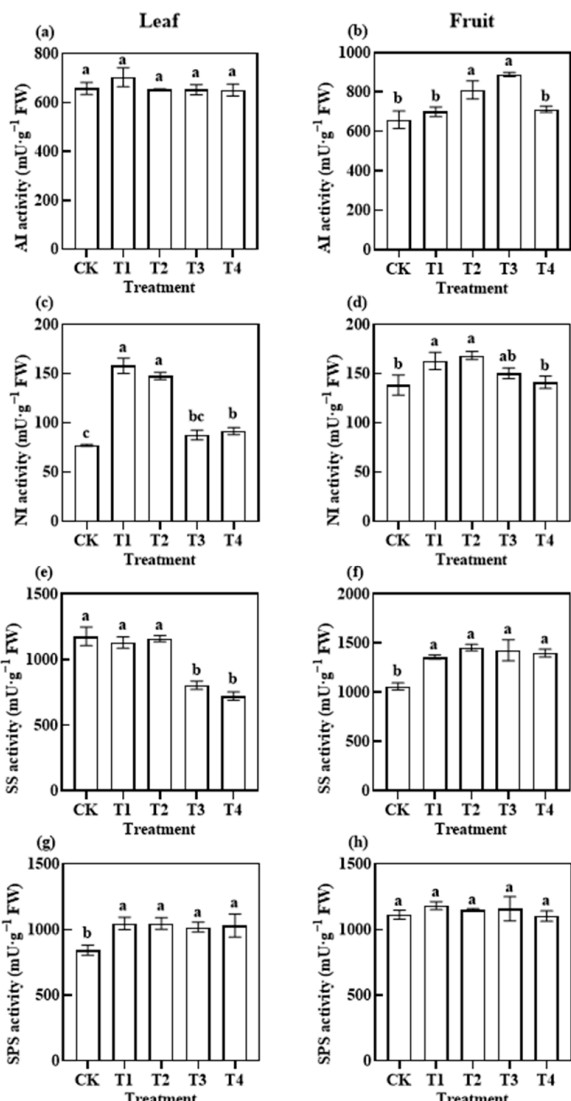

**Figure 3.** Effects of reduced fertiliser application before harvesting on the activities of C metabolism-related enzymes in plants. At the peak fruiting stage (the day of the fourth fruit harvest), samples were taken to determine the activities of C metabolism-related enzymes in leaves and fruits. AI (**a**,**b**) acid sucrose invertase; NI (**c**,**d**) neutral sucrose invertase; SS (**e**,**f**); SPS (**g**,**h**) sucrose phosphate synthase: sucrose synthase. Data were the means of three replicates with standard error (±SE). Different letters indicated significant differences between treatments at *p* of 0.05 according to Tukey's multiple range test (n = 3). With no fertiliser reduction as the control (CK), the T1, T2, T3 and T4 treatments represented 20%, 40%, 60% and 80% reduction in fertiliser application from 0 to 6 days before each harvest, respectively.

3.3.2. Photosynthetic N Metabolism-Related Enzymatic Activities

With the increase in the reduction of fertiliser dose before harvesting, the activities of N metabolism-related enzymes in leaves and fruits showed a trend of firstly increasing and then decreasing or gradually decreasing (Figure 4). T1 treatment had no significant effect on the activities of NR, NiR, GS, GOGAT and GDH in leaves. However, the GOGAT and GDH activities in leaves treated with T2 significantly increased (10.89% and 40.66%) compared with those in CK leaves. The activities of NiR and GOGAT in leaves treated with T3 and T4 were significantly lower than those in CK leaves. In fruit, T2 treatment significantly increased the activities of GS, GOGAT and GDH by 14.40%, 34.77% and 16.21%, respectively. Under T2 and T3 treatments, the activities of NiR (28.94% and 15.80%), GS (21.16% and 13.74%) and GOGAT (20.80% and 14.86%) in fruits significantly increased compared with

those in CK fruits. However, when the dose of reduced fertiliser application continued to increase (T4 treatment), the activities of N metabolism-related enzymes in the fruit showed a decreasing trend.

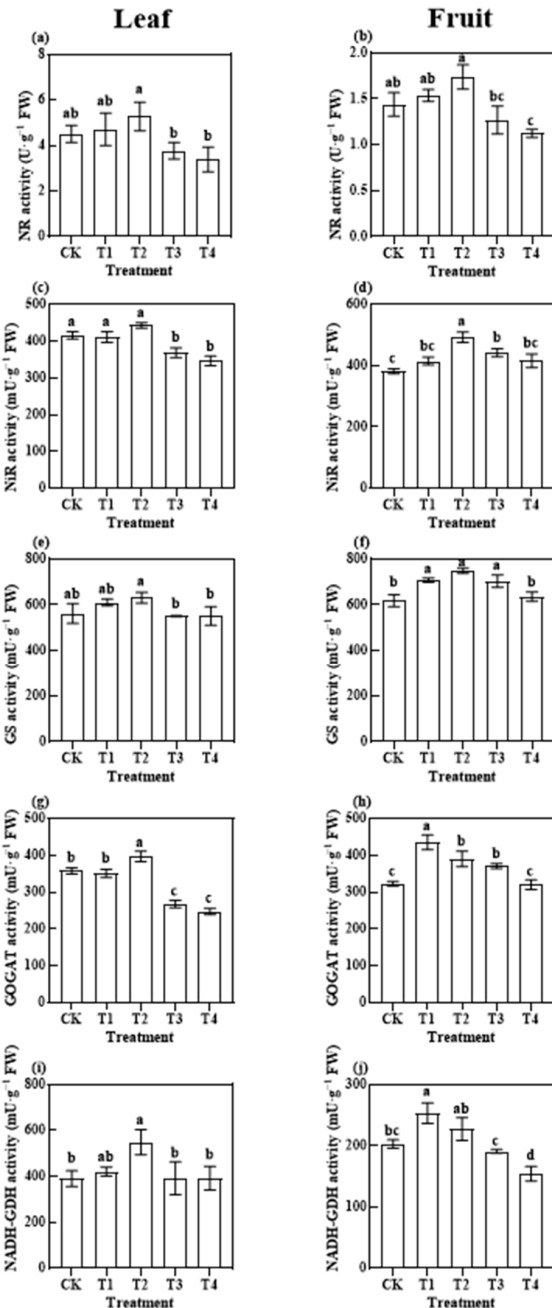

**Figure 4.** Effects of reduced fertilisation before harvest on the activities of N metabolism-related enzymes in plants. At the peak fruiting stage (the day of the fourth fruit harvest), samples were taken to determine the activities of N metabolism-related enzymes in leaves and fruits. NR (**a**,**b**) nitrate reductase; NiR (**c**,**d**) nitrite reductase; GS (**e**,**f**) glutamine synthase; GOGAT (**g**,**h**) glutamate synthase; NADH-GDH (**i**,**j**): NADH-dependent glutamate dehydrogenase. The bar size was mean ± SE (n = 3). Different letters indicated significant differences between treatments at *p* of 0.05 according to Tukey's multiple range test. With no fertiliser reduction as the control (CK), the T1, T2, T3 and T4 treatments represented 20%, 40%, 60% and 80% reduction in fertiliser application from 0 to 6 days before each harvest, respectively.

### 3.4. Responses of NUE and WUE to Fertiliser Reduction

Compared with CK, reducing the application of fertilisers before harvest had no significant effect on the NAE of N, P and K, but it gradually decreased NPE (Figure 5). The NRE of N, P, and K in T1 and T2 treatment plants was significantly higher than that in CK plants, with increases of 3.35–6.00%, 24.68–29.21% and 7.49–15.68%, respectively. However, only the NRE of P (18.99–21.85%) and K (14.79–15.93%) in T3 and T4 treatments significantly increased compared with that in CK, whereas that of N significantly decreased. Only in the T2 treatment, the NPFP of N (6.84%), P (10.12%) and K (10.14%) was significantly higher than that of the CK plants, whereas no significant difference was found between the NPFP in other treatments and in CK. Compared with the CK plants, the reduction in fertiliser application before harvest significantly increased the NHI of N (22.51–44.54%), P (29.22–51.52%) and K (23.89–35.71%) in the plants, indicating that such reduction promoted the transfer of nutrients to the fruit. Only when the pre-harvest fertilisation rate was reduced by 80% that the WUE of the plants was significantly reduced.

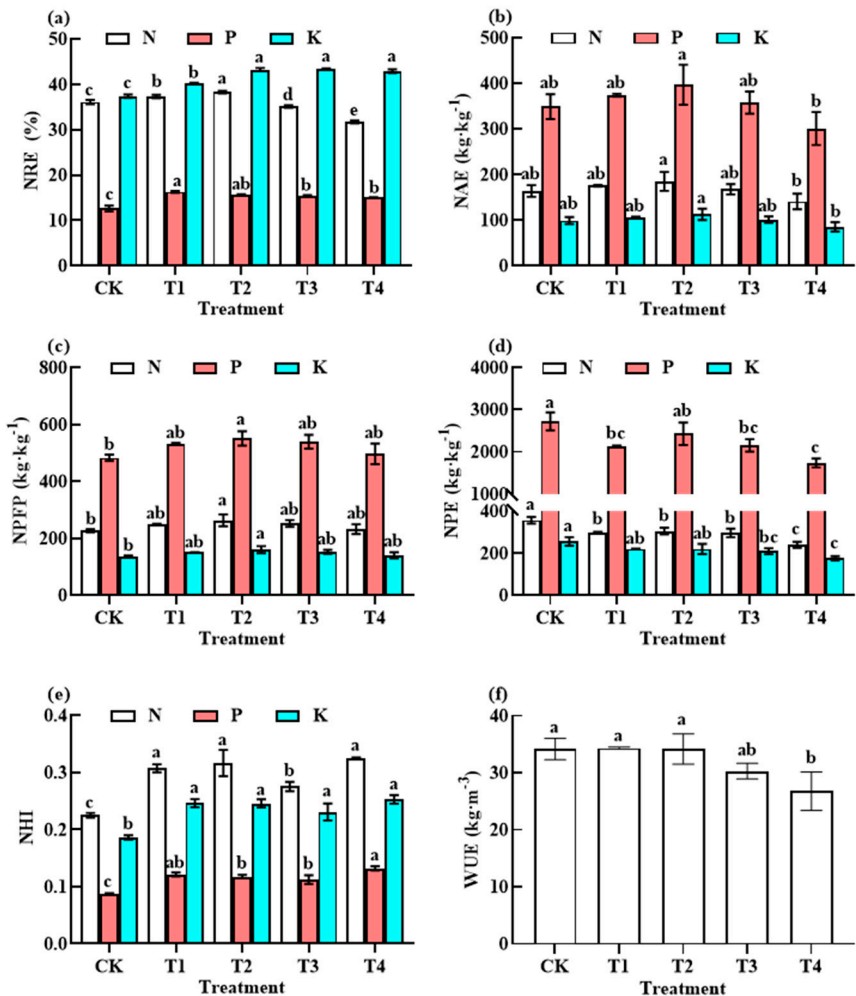

**Figure 5.** Effects of reduced fertiliser application before harvest on nutrient and water use efficiency. The amount of water irrigation and fertilisation throughout the experimental period was recorded to detect water use efficiency (WUE) and nutrient use efficiency. NAE, NRE, NPFP and NPE (**a–f**) represent the agronomic efficiency, recovery efficiency, partial-factor productivity and physiological efficiency for the applied nutrient (N, P and K), respectively. NHI represented the nutrient harvest index. Data were the means of three replicates with standard error (±SE). Different letters indicated significant differences between treatments at *p* of 0.05 according to Tukey's multiple range test (n = 3). With no fertiliser reduction as the control (CK), the T1, T2, T3 and T4 treatments represented 20%, 40%, 60% and 80% reduction in fertiliser application from 0 to 6 days before each harvest, respectively.

### 3.5. Effects of Reducing Fertiliser Application before Harvest on Fruit Quality

3.5.1. Fruit Nutritional Quality

Before pepper was harvested, a moderate reduction in nutrient input showed a certain positive effect on pepper quality indicators. However, when the nutrients were excessively reduced, the quality of the pepper significantly decreased (Table 2). When the nutrient input was reduced by 20–40% before harvesting, the contents of soluble protein, free amino acids, soluble sugar and reducing sugar were not significantly different from those in the CK fruit, in addition to the significant increase in the content of vitamin C (15.51–29.41%) and capsaicin (16.82–18.45%) in pepper. When the fertiliser application was reduced by 60–80% before harvest, the content of vitamin C, soluble protein, capsaicin and nitrate in peppers significantly decreased compared with that in CK fruits. The content of free amino acid, soluble sugar and reducing sugar content also tended to decrease.

**Table 2.** Effects of reduced fertiliser application before harvesting on the nutritional quality of peppers.

| Treatment | Vitamin C (mg·g$^{-1}$) | Soluble Protein (µg·g$^{-1}$) | Free Amino Acids (µg·g$^{-1}$) | Soluble Sugar (%) | Reducing Sugar (%) | Capsaicin (µg·g$^{-1}$) | Nitrate (µg·g$^{-1}$) |
|---|---|---|---|---|---|---|---|
| CK | 1.87 ± 0.01 c | 659.70 ± 7.59 a | 336.44 ± 14.09 ab | 8.15 ± 0.55 ab | 3.69 ± 0.13 abc | 33.65 ± 0.83 b | 235.58 ± 6.22 a |
| T1 | 2.16 ± 0.08 b | 663.16 ± 13.25 a | 339.62 ± 9.83 ab | 9.21 ± 0.15 a | 4.12 ± 0.10 ab | 39.31 ± 0.42 a | 204.28 ± 8.24 ab |
| T2 | 2.42 ± 0.02 a | 641.00 ± 10.11 a | 379.72 ± 8.75 a | 8.53 ± 0.61 ab | 4.39 ± 0.33 a | 39.86 ± 1.09 a | 185.43 ± 6.51 bc |
| T3 | 1.64 ± 0.01 d | 558.61 ± 13.05 b | 303.28 ± 16.29 bc | 6.86 ± 0.67 ab | 3.32 ± 0.31 bc | 26.12 ± 1.12 c | 162.68 ± 5.35 cd |
| T4 | 1.53 ± 0.04 d | 553.20 ± 13.04 b | 261.75 ± 14.39 c | 6.17 ± 0.42 b | 3.00 ± 0.02 c | 18.92 ± 0.88 d | 149.42 ± 7.23 d |

Data were determined on the pepper samples from the third and fourth harvests. They were the means of three replicates with standard error (±SE). Different letters indicated significant differences between treatments at *p* of 0.05 according to Tukey's multiple range test (n = 3). With no fertiliser reduction as the control (CK), the T1, T2, T3 and T4 treatments represented 20%, 40%, 60% and 80% reduction in fertiliser application from 0 to 6 days before each harvest, respectively.

3.5.2. Fruit Flavour Compounds

By using GC–MS, a total of 24 aromatic substances were detected from pepper fruits, of which 19 (CK), 22 (T1), 22 (T2), 19 (T3) and 17 (T4) were detected in the five treatments (Supplementary Table S1). With the increase in the reduction of fertiliser rate before harvesting, these aromatic substances showed a trend of gradually decreasing or firstly increasing and then decreasing (Supplementary Table S1). The random forest model was used to evaluate the importance of each substance component, and representative aromatic substances were extracted from them (Figure 6, Supplementary Figure S1 and Table S2). A total of six representative aromatic substances were extracted, namely, 3-pentanone, 2-hexenal, hexyl alcohol, 2-pentylfuran, 1-3-6-octatriene-3-7-dimethyl- and propanoic acid-2-methyl-4-methylpentyl ester (Figure 6, Table 3 and Supplementary Table S2). The model interpretation rate of random forest analysis was 95%, indicating that the predicted aromatic substances had a great fit with the actual aromatic substances (Supplementary Figure S1). Amongst the above six kinds of aromatic substances, the contents of 3-pentanone, 2-pentylfuran and propanoic acid-2-methyl-4-methylpentyl ester all showed a decreasing trend with the increase in the reduction of fertiliser rate. However, a 20% reduction in fertiliser application before harvest significantly induced an increase in the contents of 2-hexenal (61.15%) and 1,3,6-octatriene-3-7-dimethyl- (42.92%) in the fruit compared with CK. When the reduced application rate increased to 40%, the 2-hexenal content in the fruit (43.06%) was significantly higher than that in the CK fruit. On the whole, reducing the application of fertiliser by 20–40% before harvesting induced a significant increase in the total amount of representative aromatic substances by 47.35–31.76%.

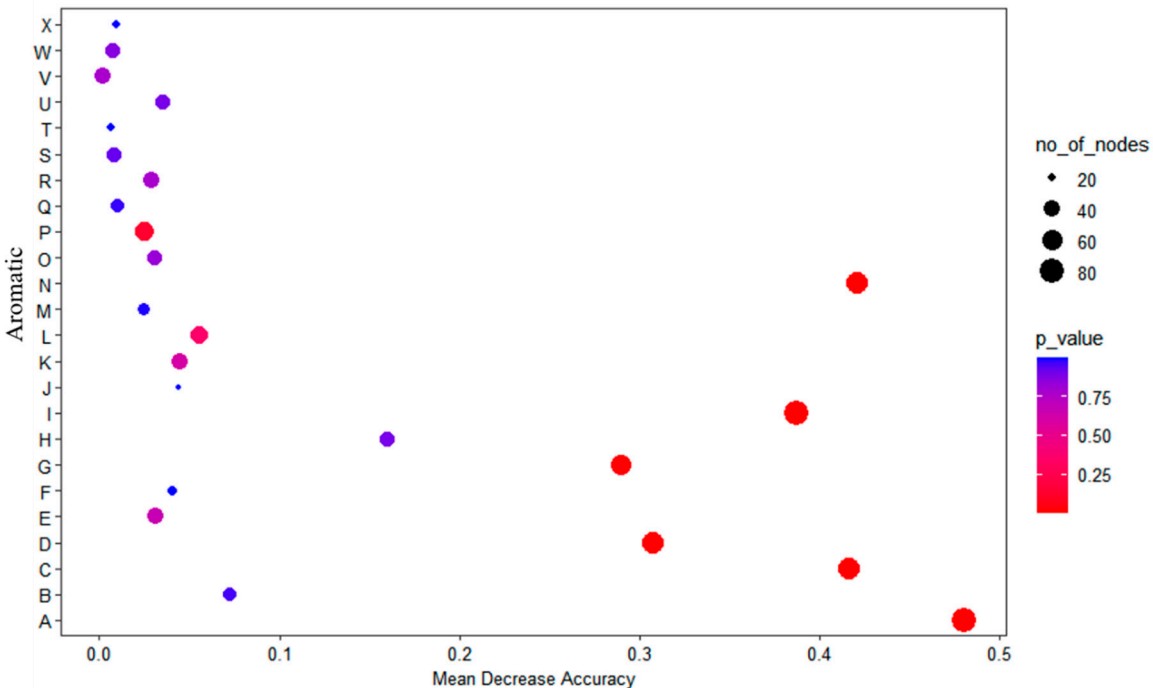

**Figure 6.** Random forest model for 24 aromatic substances. The A–X in the figure represents aromatic substances. The detailed data are shown in Supplementary Table S1. The size of the horizontal axis represented the number of mse-increase.

**Table 3.** Effects of reduced fertiliser application before harvest on aromatic compounds in pepper fruits.

| Aromatic Content ($\mu g \cdot kg^{-1}$) | Treatment | | | | |
|---|---|---|---|---|---|
| | **CK** | **T1** | **T2** | **T3** | **T4** |
| 3-Pentanone | 39.14 ± 5.00 a | 36.34 ± 1.45 a | 15.74 ± 1.13 b | ND | ND |
| 2-Hexenal | 5086.10 ± 253.62 b | 8196.24 ± 407.08 a | 7276.33 ± 288.93 a | 3313.78 ± 4.37 c | 2768.89 ± 228.06 c |
| Hexyl alcohol | 2467.36 ± 144.46 a | 3164.41 ± 177.75 a | 3181.07 ± 462.69 a | 1152.88 ± 3.07 b | 1240.03 ± 55.70 b |
| 2-Pentylfuran | 145.56 ± 11.78 a | 53.39 ± 4.20 bc | 69.85 ± 6.78 b | 28.41 ± 0.04 c | ND |
| 1,3,6-Octatriene, 3, 7-dimethyl- | 545.72 ± 4.41 b | 779.97 ± 3.93 a | 428.00 ± 23.31 c | 274.46 ± 0.36 d | 118.03 ± 3.90 e |
| Propanoic acid, 2-methyl-, 4-methylpentyl ester | 51.05 ± 2.06 a | 51.45 ± 3.84 a | 11.15 ± 1.46 b | 10.67 ± 0.01 b | 19.99 ± 0.24 b |
| Total | 8334.93 ± 85.91 b | 12281.80 ± 242.74 a | 10982.15 ± 779.11 a | 4780.20 ± 6.85 c | 4146.94 ± 207.15 c |

The data were determined on the pepper samples from the third and fourth harvests. Data was the means of three replicates with standard error (±SE). Different letters indicated significant differences between treatments at *p* of 0.05 according to Tukey's multiple range test (n = 3). Taking no fertiliser reduction as the control (CK), the T1, T2, T3, and T4 treatments represented 20%, 40%, 60% and 80% reduction in fertiliser application from 0 to 6 days before each harvest, respectively.

### 3.6. Effects of Reduced Fertiliser Application before Harvest on Pepper Yield

With the decrease in fertiliser application dose before harvest, the pepper yield showed a trend of firstly being stable and then decreasing (Figure 7). Compared with CK, reducing the amount of fertiliser applied by 20–40% before harvest had no significant effect on the pepper yield. When the application rate was reduced by 60%, the pepper yield began to show a downward trend, but no significant effect was found compared with that of CK plants. When the fertiliser dose was further reduced to 80%, the pepper yield significantly decreased by 21.62% compared with that of CK plants.

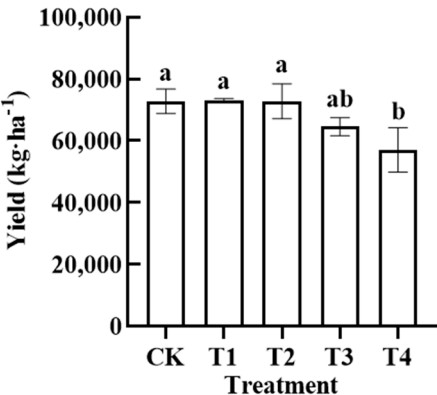

**Figure 7.** Effect of reducing the application of different doses of fertiliser before harvesting on the yield of pepper. The yield of peppers represented the total yield from six harvests during the experimental period. Different letters indicated significant differences between treatments at *p* of 0.05 according to Tukey's multiple range test (n = 3). With no fertiliser reduction as the control (CK), the T1, T2, T3 and T4 treatments represented 20%, 40%, 60% and 80% reduction in fertiliser application from 0 to 6 days before each harvest, respectively.

## 4. Discussion

### 4.1. Effects of Reduced Fertiliser Rate on Growth and Biomass before Harvest

Suitable fertiliser supply has a remarkable positive effect on plant growth and yield, leading to increased total aboveground dry matter [38,39]. Numerous studies tended to explore the effects of inputting different rates of nutrients on plant growth and production during the whole growth period [24,26]. They focused on obtaining an optimal fertilisation scheme for crops from the perspective that crops face the same nutrient status at all times throughout the growth period, often ignoring the dynamic changes in plant demand for nutrients in different growth periods. Therefore, the design of the experimental content in the present study comprehensively considered the characteristics of different nutrient dose demands of plants in different growth stages and the objective fact that the demand for nutrients before harvesting tends to decrease [25]. The experimental results of plant growth and dry-matter accumulation showed that the plant height and stem diameter of peppers decreased with the increase in the reduction of fertiliser doses from the first day to the sixth day before each harvest of peppers (Figure 1a,b). On the 40th day after treatment, the plant height of plants with a 40–80% reduction in fertiliser dose was significantly lower than that of the CK plants (Figure 1a). The results showed that moderate reductions in fertiliser application prior to harvest did not negatively affect plant growth and suggested a threshold for reduced application rates. Dry-matter accumulation directly reflects the physiological and metabolic efficiency of plants [40]. Within the scope of the experimental design, with the increase of the reduced fertiliser dose before harvesting, the dry-matter accumulation in roots, stems and leaves of pepper plants was not significantly different from that of CK plants. However, the dry-matter accumulation of fruit was significantly lower than that of CK plants after reducing the application rate by more than 40%. Meanwhile, as the proportion of fruit in different plant tissues was the highest and the biomass of other tissues did not tend to decrease with the increase in reduced application rate, the variation trend of dry-matter accumulation in different plant organs was in line with the 'source-sink' theory of plants [41]. When the nutrient level of plants gradually decreases, the photosynthetic metabolic process may have a decreasing trend. In this case, the energy produced by the plant was preferentially used for the growth and metabolism of the 'source' tissue, and the transport to the 'sink' was reduced. Therefore, with the increase in nutrient reduction application rate before harvest, the biomass of roots, stems and leaves, which guaranteed the functions of plant nutrient absorption, transport and metabolism, had no significant decrease trend, whereas the accumulation of dry matter in fruits as 'sinks' gradually decreased.

*4.2. Effects of Reduced Application of Different Fertiliser Rates on Photosynthetic Metabolism before Harvest*

For plants, photosynthesis provides material and energy for plant growth, and it is the basis for the formation of crop yields [42]. Most of the energy and its intermediates produced by plant photosynthesis are used for C and N metabolism, providing energy and substrates for other physiological and biochemical processes in plants [43]. The metabolic processes of C and N are closely linked because they must share organic C and energy provided directly by photosynthetic electron transfer and $CO_2$ fixation or that provided by respiration via glycolysis, tricarboxylic acid cycle and mitochondrial electron transport chain pathways [44,45]. These two metabolic processes are relatively independent. On the whole, the photosynthetic C metabolism process mainly provides energy and C skeleton for N metabolism, whilst N metabolism independently performs amino acid synthesis on the basis of reducing power and intermediate products, such as ATP and NAD(P)H, generated by light reaction with C metabolism [46]. In the process of N metabolism, the GS/GOGAT cycle, as the main pathway of N assimilation in higher plants, plays a key role in the conversion of inorganic N to organic N [47]. In the present study, the nutrient physiological metabolism level and photosynthetic metabolism capacity of plants were measured to verify the reason behind the difference in biomass. The measurement results of N-metabolising enzyme activities in leaves and fruits showed that mildly reducing fertiliser (20–40%) before harvest had no significant effect on the activities of NR, NIR and GS in pepper leaves, whereas 40% reduction in application rate promoted a significant increase in the GOGAT and GDH activities in leaves (Figure 4). However, when the reduction in application rate was increased to 60–80%, the activities of NIR and GOGAT in leaves decreased significantly compared with those in CK plants and the activities of NR, GS and GDH also decreased (Figure 4). The results indicated that the leaf N metabolism process was affected by the level of nutrient supply before harvesting, and the appropriate reduction in nutrient input before harvesting could not have a negative effect on the N metabolism process. The changing trend of N-metabolising enzyme activities in pepper fruits under different treatments was basically the same as that in leaves, but the difference was that the reduction in fertiliser supply by 20–60% before harvesting did not cause a reduction in the N metabolism-related enzyme activities in fruits (Figure 4). At the rate of 60% fertiliser reduction, the NIR and GOGAT activities of the fruit were even significantly higher than those of the CK plants. Under 80% fertiliser reduction treatment, only the activities of NR and GDH were significantly lower than those of the CK plant; the activities of NIR, GS and GOGAT were not significantly different from those in the CK plant (Figure 4). The changing trend of N-metabolising enzyme activities in leaves and fruits is basically consistent with the previous research results [48–50]. That is, providing plants with excess nutrients during a specific growth period could cause excessive accumulation of nutrients in the rhizosphere, thus affecting the plant's absorption and transport of elements and then causing the photosynthetic metabolism-related enzyme activities in the aerial part of the plant to decrease. Appropriately reducing the supply of nutrients could make the rhizosphere nutrient environment of the plant at a more suitable level, and the coordination of vegetative growth and reproductive growth could be strengthened, which is conducive to promoting the physiological and metabolic process of the plant.

In the process of plant C metabolism, sucrose, as the most important metabolite of plants, is not only the main form of plant assimilation products transported from 'source' to 'storage' but also an important storage form of sugar in mature fruits and storage organs [51]. In the process of sucrose metabolism, SPS is the key rate-limiting enzyme controlling sucrose synthesis, and SS could simultaneously regulate the synthesis and decomposition of sucrose. Both are involved in coordinating the distribution of fixed C and sucrose in leaves and fruits [52]. Sucrose invertase irreversibly catalyses the hydrolysis of sucrose to glucose and fructose and plays an important role in sugar transport, processing and storage. In accordance with the optimal pH for the enzymatic reaction, sucrose invertase is divided into two forms: acid invertase and neutral invertase [53,54]. Acid invertase is mainly

found in vacuoles (soluble) or bound to the cell wall (insoluble) [55,56]. The former mainly catalyses the hydrolysis of sucrose into hexose and regulates the accumulation of sugar in plant tissues and the utilisation of sucrose in vacuoles. The latter is considered to be the key enzyme regulating sucrose unloading, and it plays an important role in maintaining sucrose transport between 'source' and 'sink' tissues. Neutral invertase mainly accumulates in the cytoplasm, and its role is mainly to indirectly regulate sucrose metabolism by regulating the level of intracellular hexose [57]. The results of the determination of sucrose metabolism-related enzyme activities in the present study showed that a 20–40% reduction in fertilisation dose before harvest significantly increased the NI activity in leaves (Figure 3). The fertilisers reduced by 20–80% significantly induced the increase of SPS activity in leaves, but it had no significant effect (20–40%) on the SS activity or significantly decreased it (60–80%, Figure 3). The results indicated that reducing fertiliser application before harvest promoted not only the synthesis of sucrose in leaves but also its conversion to hexose and its transport to the 'sink' tissue. This conversion or transport process may be more sensitive to reduced fertiliser application. In fruit, reducing fertiliser application before harvest significantly increased the activity of SS but had no significant effect on the activity of SPS (Figure 3). The reason may be that leaves are the 'source' organs of plants, and the main use of sucrose synthesised by photosynthesis tends to be transported to the 'sink'; meanwhile, in the 'sink' of fruits, the synthesis of sucrose decreased, and the decomposition of sucrose improved [58]. The decomposed sucrose has two main purposes: to synthesise storage substances, such as starch, and to convert into other substances, such as fructose and glucose, to improve fruit quality [59].

*4.3. Effects of Reducing the Application of Different Fertiliser Rates before Harvesting on Nutrient Utilisation*

When the input amount of fertiliser is greater than its actual demand, it could fail to increase not only the plant's absorption and utilisation efficiency of elements but also its yield [59]. At this time, an appropriate reduction of fertiliser supply could improve the plant's absorption and utilisation efficiency of elements [49]. The accumulation of N, P and K elements during the experiment was measured, and NUE was calculated to explore the effect of nutrient reduction before harvesting on the absorption and utilisation of elements and verify the experimental results related to the C and N metabolism of plants (Figures 2b and 5). During the whole experimental period, the accumulation of N, P and K in plants showed a trend of firstly increasing and then decreasing with the increase in chemical fertilisers (Figure 2b). The accumulation of N and K in plants with a 20–40% reduction in fertiliser before harvesting was significantly higher than that in the CK plants. The accumulation of P only significantly increased in plants with a reduction of 20%. However, when the rate of fertiliser reduction continued to increase (60–80%), no significant difference was observed in the accumulation of N, P and K between the plants and the CK plants. This result was also in line with the experimental assumption, that is, in the fruit colour change period before pepper harvesting, the plant's demand for nutrients tends to decrease compared with that in the fruit setting period or the fruit enlargement period. Therefore, appropriately reducing the fertiliser supply during the fruit colour-changing period not only did not cause a reduction in nutrient absorption and accumulation by plants but also promoted the absorption and utilisation of elements by plants.

Previous research results have shown that excessive fertiliser input could lead to reduced NUE [60]. Amongst the indicators for evaluating element utilisation efficiency, NRE and NAE are mainly used to characterise the absorption efficiency per unit of nutrient, and the yield increase caused by a per applied unit of nutrient, respectively [61]. Compared with CK, the fertilisers reduced by 20–80% in 1–6 days before harvest had no significant effect on NAE (Figure 5). The results indicated that excessive nutrient input before harvest could not lead to a significant increase in yield. When the fertiliser was reduced by 20–40% before harvest, the recovery efficiency of N, P and K significantly improved. The recovery efficiency of P and K remained significantly higher than that in the CK plants when the

dose of fertiliser reduction continued to increase to 80% (Figure 5). The reason for this result may be that the plant's demand for nutrients decreased during the colour-changing period of pepper and the higher nutrient content in the rhizosphere at this time easily limited the plant's ability to absorb nutrients. At this time, appropriately reducing the nutrient supply of plants was more conducive to the absorption, transformation and accumulation of nutrients by plants. However, due to the differences in the demand of plants for different elements, the NRE calculated by different elements also differed. NPFP and NPE are mainly used to characterise the fruit yield obtained by increasing the unit nutrient input and the increase in fruit yield corresponding to the unit nutrient accumulation of the plant, respectively [37]. Both are used to characterise the corresponding relationship between nutrient input and yield output. In this experiment, only the partial factor productivity of N, P and K in plants with 40% fertiliser reduction was significantly higher than that in CK plants (Figure 5), indicating that the highest yield per unit nutrient input was obtained when fertiliser was reduced by 40%. Meanwhile, the trend of NHI results showed that the reduction in fertiliser application significantly improved the harvest index of N, P and K (Figure 5), also proving that the effect of nutrients on plant yield or the nutrient demand of plants decreased during the fruit colouring stage. Therefore, the magnitude of the decline in production caused by reduced nutrient supply demonstrated a downward trend. However, when the influence of the nutrient content of the organic substrate itself on the experimental results was excluded, the physiological efficiency of N, P and K in the plants after reducing fertiliser application was found to be decreased compared with the CK plants (Figure 5). The reason for this result is that reducing fertiliser application before harvesting promoted the absorption and accumulation of nutrients, but it did not cause a significant increase in crop yield (Figure 7). Therefore, the trend of NPE, NPFP and NHI determination results was inconsistent. The finding also showed that when evaluating the influence of exogenous nutrient input on plant yield, the influence of the nutrient content of the rhizosphere environment itself on the test results should also be considered.

### 4.4. Effects of Reducing the Application of Different Fertiliser Rates before Harvesting on Yield and Quality

In this experiment, a 20–40% reduction in fertiliser application 1–6 days before harvesting did not cause a significant decrease in pepper yield. When the fertiliser application was reduced by 60%, the pepper yield began to decline and to continue to increase the reduction could lead to a significant decrease in pepper yield (Figure 7). The changing trend of yield results was also basically consistent with that of plant physiological indicators, biomass and element accumulation. In particular, reducing the application of low-dose fertiliser before harvesting had no significant effect on the physiological metabolism, growth and element utilisation of peppers or had a promoting effect. However, excessively reducing the fertiliser application could have a negative effect on the physiological metabolism or growth process of the plant.

Capsaicin and aromatic substances are considered important indicators for determining the flavour of peppers [18]. Only reducing the application to 20–40% before harvesting significantly increased the vitamin C and capsaicin content of peppers (Table 2). However, reducing such applications before harvest had no significant effect on other nutritional quality indicators or significantly reduced the quality. With the increase in the reduction of application rate, the quality of pepper showed a downward trend. Compared with those in CK plants, the contents of the 24 kinds of aromatic compounds detected in pepper fruits, except for 2-pentylfuran, 3-pentanone, eicosane and 10-methyl-, showed a trend of firstly increasing and then decreasing with the increase in the reduction of fertiliser application before harvest; the maximum value generally appeared in the 20% or 40% treatments (Supplementary Table S1). The changing trend of the total content of six representative aromatic compounds screened by the random forest model was the same as the above results. The experimental results are also consistent with the previous conclusions that rhizosphere nutrient content affects the content of aromatic fruit compounds [62]. However,

no unified conclusion could be made about the effect of different nutrient levels on the content of aromatic substances in plants. The reasons may be related to species and the optimal nutrient level required by plants. The formation of aromatic compounds requires C and N metabolism to provide substrates [63]. In the present work, fertilisers with a 20–40% reduction before harvesting had a higher content of aromatic compounds, and the trend was basically consistent with that of physiological metabolism related to C and N metabolism (Figures 3 and 4 and Table 3). The reason may be that the physiological, metabolic process is better when the plant is at an appropriate nutrient level. However, the change in specific substance content should also be analysed in conjunction with the physiological and metabolic pathways.

## 5. Conclusions

During the fruit colour-changing period 1–6 days before harvest, a risk of excessive fertilisation could be present if fertilisation is conducted in accordance with the conventional rate. The fertilisation amount reduced by 20–40% before harvest could promote the absorption and utilisation of elements by plants. It could also stabilise or promote C and N metabolism and ultimately improve fruit quality. The reduced rate did not negatively affect the plant biomass and yield, for increased economic and ecological benefits and to comprehensively consider the effects of different fertilisation rates on plant growth and production before harvesting, reducing the fertilisation rate by 40% from days 6 to 0 before fruit harvesting is recommended. The experimental results are beneficial to improve the economic benefits of agricultural producers on the basis of improving the ecological environment.

**Supplementary Materials:** The following are available online at https://www.mdpi.com/article/10.3390/agronomy12123004/s1, Figure S1: Relations between measures of importance and between rankings according to different measures. Table S1: Effects of reducing different doses of fertilisers before harvest on aromatic compounds in pepper fruit. Table S2: Screening of representative aromatic substances based on random forest model.

**Author Contributions:** J.W.: Conceptualisation, Investigation, Conduct experiments, Data curation, Methodology and Writing—original draft. Z.G.: Writing—review, Editing, Conducting experiments, Data curation. T.S.: Methodology, Conduct experiments, Funding acquisition. W.H.: Conduct experiments, Resources. Y.J.: Data Measurement and Analysis. X.H.: Funding acquisition, Project administration, Supervision and Writing—review and editing. X.L. and Z.Z.: Experimental design and Writing—review. All authors have read and agreed to the published version of the manuscript.

**Funding:** This work was supported by the Key Research and Development Program of Shaanxi Province in China (2022NY-116, 2022ZDLNY03-11), China Agriculture Research System (CARS-23-D06), and the Integration and Promotion of Agricultural Science and Technology Innovation in Shaanxi Province (NYKJ-2022-YL(XN)06).

**Data Availability Statement:** Data is contained within the article or supplementary material. The data presented in this study are available in [supplementary material].

**Conflicts of Interest:** The authors declare that they have no known competing financial interests or personal relationships that could have appeared to influence the work reported in this paper.

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
