# Peer review of "Preharvest Reduction in Nutrient Solution Supply of Pepper (Capsicum annuum L.) Contributes to Improve Fruit Quality and Fertilizer Efficiency While Stabilising Yields"

_agronomy, doi:10.3390/agronomy12123004_

Round 1
Reviewer 1 Report
The manuscript is very interesting, from the perspective of reducing chemical inputs and environmental sustainability. I suggest removing lines 115-116 from the introduction and inserting them in the conclusion, expanding on them; I also suggest following the author's guidelines for citing the bibliography.
Author Response
Replies to Editor and Reviewers
Response to Reviewer 1 Comments:
Point 1: The manuscript is very interesting, from the perspective of reducing chemical inputs and environmental sustainability. I suggest removing lines 115-116 from the introduction and inserting them in the conclusion, expanding on them; I also suggest following the author's guidelines for citing the bibliography.
Answer: Thank you for your recognition of our work and for spending a lot of time revising this manuscript. We checked the references throughout the manuscript to ensure the accuracy of the cited references. Meanwhile, we follow your suggestion and move lines 115-116 of introduction to the conclusion. Finally, we checked the raw data of the full text and the data in the manuscript and confirmed that all the data in the article were accurate. We tried our best to improve the manuscript. Finally, thanks once again to the Editors and Reviewers for giving us such valuable comments and suggestions.

Reviewer 2 Report
Dear Authors
I think you can change (improve) the manuscript tittle.
Line 22 in abstract is not clear. rewrite it.
For a reader who reads the abstract before reading the document, it is complicated to understand. Rewrite the abstract, also attention to the abstract limition words.
introduction should be more precise and specifically address the aim and the topic of reduced fertilizer before harvest. Please find and mention some published papers regarding reduced fertilizer 6 days or a few days before harvesting.
In Table 1, I understand how you applied the treatment, not inside the text. Now my question is, how can the grower follow along like this? Is it really practical? Also I suggest to change (Write) them to 15-7 days and 6 days to 0 days before each harvest.
other question, how do you decide to apply this amount of water? why last three harvesting periods need more water? please refernce to some paper, book....
please describe DAF0, DAF10, .. in figure 1 caption.
You measured many good factors and did a great job. while the presentation of your data is not really good. Before accepting and publishing your paper, you need a significant edition in text, table, and figure. emphasize the significance of this research in the introduction and goal.
As I understand it, in the conclusion, line 676, "reducing fertilizer from day 1 to day 6 before harvesting" has the opposite meaning of what you did. (You reduced the fertilzer from day 6 before harvesting).
Author Response
Response to Reviewer 2 Comments:
Point 1: I think you can change (improve) the manuscript tittle.
Answer: Thank you for your recognition of our work and for spending a lot of time revising this manuscript. We have followed your suggestion and revised the title of this manuscript to "Preharvest reduction in nutrient solution supply of pepper (Capsicum annuum L.) contributes to improve fruit quality and fertilizer efficiency while stabilizing yields".
Point 2: Line 22 in abstract is not clear. rewrite it.
Answer: Thanks for your suggestion. We reworked the statement in line 22.
Point 3: For a reader who reads the abstract before reading the document, it is complicated to understand. Rewrite the abstract, also attention to the abstract limition words.
Answer: Thanks for your suggestion. Some of the descriptions in the abstract may be difficult to understand, and we have rewritten and double-checked the relevant content.
Point 4: Introduction should be more precise and specifically address the aim and the topic of reduced fertilizer before harvest. Please find and mention some published papers regarding reduced fertilizer 6 days or a few days before harvesting.
Answer: Thanks for your suggestion. In previous studies, scholars on precision fertilization have mostly calculated fertilizer dosage reduction based on the total fertilizer application during the whole reproductive period, rather than considering precision fertilization strategies based on the variability of water and fertilizer requirements at different growth stages. At the same time, it has been shown that during the pre-harvest period of the fruit, there was a decreasing trend in plant nutrient requirements. The growth stage corresponding to the 6 days before harvest of pepper is dominated by the color change of the fruit. And previous research by our group found that reducing fertilizer application within 6 days before harvesting peppers stabilized yield and improved fruit quality. We red-flagged the above research descriptions and their references in the introduction. In the context of the above-mentioned research, we would like to explore strategies for precision fertilization from the perspective of specific growth stages based on the reduced nutrient requirements of crops before harvesting.
Point 5: In Table 1, I understand how you applied the treatment, not inside the text. Now my question is, how can the grower follow along like this? Is it really practical? Also I suggest to change (Write) them to 15-7 days and 6 days to 0 days before each harvest.
Answer: Thanks for your suggestion. We have followed your suggestion and revised the description in the article (15-7 days and 6 days to 0 days), which will help the reader better understand the article. In the actual production process of Chinese facility agriculture, especially in substrate cultivation production, the frequency of fertilizer application by farmers is basically once every 2-3 days. Therefore, reducing the fertilizer application 6-0 days before each harvest according to our findings will not only not reduce the yield of peppers, but also help to improve the quality of peppers and improve the environment, and the results of the study are in line with the actual production needs.
Point 6: other question, how do you decide to apply this amount of water? why last three harvesting periods need more water? please refernce to some paper, book....
Answer: Thanks for your suggestion. Our determination of the amount of plant irrigation and fertilization depends on a pre-experiment on the relationship between plant growth and irrigation in the organic matter cultivation model. A relative water content of 60% favored plant growth in pots of 6.5 m × 1.2 m size and corresponding experimental environmental parameters. For the difference between the first three (pre-growth) and the last three (mid-late growth) fertilizer applications, we based it on previous studies that the nutrient requirement in the early growth stage of the crop is less than that in the fruiting stage and the later stage, on the other hand, because in actual production, the plants are relatively weak and have low nutrient requirements in the early fruiting stage of pepper, while in the fruiting stage, the yield rises and more nutrients are needed for growth requirements. We have followed your suggestion and added the references in the corresponding place in the article.
Point 7: please describe DAF0, DAF10, .. in figure 1 caption
Answer: Thanks for your suggestion. We added a relevant explanation in the annotation of Figure 1.
Point 8: You measured many good factors and did a great job. while the presentation of your data is not really good. Before accepting and publishing your paper, you need a significant edition in text, table, and figure. emphasize the significance of this research in the introduction and goal.
Answer: Thank you for recognizing our work. We followed your suggestion and rechecked the entire manuscript to ensure the accuracy of the experimental data. The revised manuscript has been edited and proofread by a professional English editing company, ShineWrite, had a certified USA business account. Thank you for your recognition of our work and for spending a lot of time revising this manuscript.

Reviewer 3 Report
Dear Authors,
Your research work is interesting because it shows an effective strategy for fertilizer efficiency, yield stabilization and pepper fruit quality improvement. The subject matter is very important nowadays. Ways to increase yields, improve the quality of the raw material and protect the environment should be sought. I agree with the You, that these results have important implications for institutional precision fertilization programs and the improvement of agroecological environment
This paper is prepared in the usual way for scientific work. However, I have a few comments and remarks. Some are debatable.
1. In the text, reference numbers should be placed in square brackets [ ], and placed before the punctuation; for example [1], [1–3] or [1,3]. Prepare a document according to the guidelines for Agronomy https://www.mdpi.com/journal/agronomy/instructions.
2. Abstract. Lines 31-37. Please rewrite these sentences. I don't understand their meaning.
3. Introduction. Lines 115-116. In my opinion, this sentence should be in conclusion.
4. Flavour quality of fruit. Line 224. And Fertiliser use efficiency (FUE) and WUE. Lines 233-238. Use the Microsoft Equation Editor or MathType Equation Add-in. Equations should be editable by the editorial office and not appear in a picture format.
5. Conclusion. Too long introduction (first sentence). It brings nothing. Please delete or rewrite.
In my opinion, the results and discussion chapters are well written.
The language appears to be correct, but I don't feel qualified to judge about the English language and style.
I recommend for publication in Agronomy after the indicated corrections.
Good luck!
Sincerely yours
Reviewer
Author Response
Response to Reviewer 3 Comments:
Point 1: Your research work is interesting because it shows an effective strategy for fertilizer efficiency, yield stabilization and pepper fruit quality improvement. The subject matter is very important nowadays. Ways to increase yields, improve the quality of the raw material and protect the environment should be sought. I agree with the You, that these results have important implications for institutional precision fertilization programs and the improvement of agroecological environment.
Answer: Thank you for recognizing our work.
Point 2: In the text, reference numbers should be placed in square brackets [ ], and placed before the punctuation; for example [1], [1–3] or [1,3]. Prepare a document according to the guidelines for Agronomy https://www.mdpi.com/journal/agronomy/instructions.
Answer: Thanks for your suggestion. We have reformatted the references in the full text according to guidelines for Agronomy.
Point 3: Abstract. Lines 31-37. Please rewrite these sentences. I don't understand their meaning.
Answer: Thanks for your suggestion. We have rewritten the section for your review.
Point 4: Introduction. Lines 115-116. In my opinion, this sentence should be in conclusion.
Answer: Thanks for your suggestion. We follow your suggestion and move lines 115-116 of introduction to the conclusion. And we checked the raw data of the full text and the data in the manuscript and confirmed that all the data in the article were accurate.
Point 5: Flavour quality of fruit. Line 224. And Fertiliser use efficiency (FUE) and WUE. Lines 233-238. Use the Microsoft Equation Editor or MathType Equation Add-in. Equations should be editable by the editorial office and not appear in a picture format.
Answer: Thanks for your suggestion. We modified the representation of the formula based on your suggestion.
Point 6: Conclusion. Too long introduction (first sentence). It brings nothing. Please delete or rewrite.
Answer: Thanks for your suggestion. We have removed this section based on your suggestion.
Point 7: In my opinion, the results and discussion chapters are well written.
Answer: Thank you for recognizing our work. We tried our best to improve the manuscript. Finally, thanks once again to the Editors and Reviewers for giving us such valuable comments and suggestions. Thank you for your recognition of our work and for spending a lot of time revising this manuscript.

Round 2
Reviewer 2 Report
Dear Authors
Thanks for your edition and comment.